# Plasma Dephosphorylated-Uncarboxylated Matrix Gla-Protein in Systemic Sclerosis Patients: Biomarker Potential for Vascular Calcification and Inflammation

**DOI:** 10.3390/diagnostics13233526

**Published:** 2023-11-24

**Authors:** Judith Potjewijd, Rachid Tobal, Karin A. Boomars, Vanessa V. P. M. van Empel, Femke de Vries, Jan G. M. C. Damoiseaux, Leon J. Schurgers, Pieter van Paassen

**Affiliations:** 1Department of Internal Medicine, Division Clinical and Experimental Immunology, Maastricht University Medical Center, 6229 HX Maastricht, The Netherlands; rachid.tobal@mumc.nl (R.T.); p.vanpaassen@maastrichtuniversity.nl (P.v.P.); 2Department of Respiratory Medicine, Erasmus MC, University Medical Center Rotterdam, 3015 GD Rotterdam, The Netherlands; k.boomars@erasmusmc.nl; 3Department of Cardiology, Maastricht University Medical Center, 6229 HX Maastricht, The Netherlands; vanessa.van.empel@mumc.nl; 4Department of Biochemistry, Cardiovascular Research Institute Maastricht, Maastricht University, 6211 LK Maastricht, The Netherlands; femke.devries@maastrichtuniversity.nl (F.d.V.); l.schurgers@maastrichtuniversity.nl (L.J.S.); 5Central Diagnostic Laboratory, Maastricht University Medical Center, 6229 HX Maastricht, The Netherlands; jan.damoiseaux@mumc.nl

**Keywords:** systemic sclerosis, Matrix Gla Protein, cardiovascular risk, biomarker

## Abstract

Background: Systemic sclerosis (SSc) patients face an elevated risk of cardiovascular disease (CVD), even when classic cardiovascular risk factors are considered. Plasma dephosphorylated-uncarboxylated Matrix Gla-protein (dp-ucMGP), an inactive form of MGP, is associated with increased CVD risk. Smooth muscle cells, implicated in SSc’s development, are the primary dp-ucMGP producers. This study assessed dp-ucMGP levels and initial CVD events in early-diagnosed SSc patients, investigating its potential as a CVD and all-cause mortality predictor over time. Methods: In a cohort of 87 SSc patients (excluding those with pre-existing CVD or on dialysis), baseline dp-ucMGP levels were measured, along with cardiovascular risk factors. Validation involved assessing dp-ucMGP in a subset of treatment-naive SSc patients. Results: A significantly elevated median dp-ucMGP level of 634 pmol/L (IQR 301) compared with healthy controls (dp-ucMGP < 393 pmol/L; *p* < 0.001) was observed. Validation in a treatment-naive SSc patient subset yielded similar results (median 589 pmol/L; IQR 370). During a median 10.5-year follow-up among 78 SSc patients, 33.3% experienced their first CVD event, independent of traditional risk factors. Elevated dp-ucMGP levels (>634 pmol/L) correlated with a higher risk of CVD and/or death (log-rank test: *p* < 0.01). Conclusions: In summary, dp-ucMGP emerges as a novel biomarker in SSc patients, with elevated levels indicating an increased risk of CVD and/or mortality in this population.

## 1. Introduction

Matrix Gla-protein (MGP) is a vitamin K-dependent protein involved in the inhibition of both intimal and medial vascular calcification. It is a small secretory protein (14 kD) that is primarily secreted by vascular smooth muscle cells (VSMCs) and endothelial cells (ECs) in the arterial wall [1]. MGP requires vitamin K-dependent carboxylation as well as phosphorylation to become activated and to obtain extra negative charge, facilitating binding to calcium salts with high affinity, thereby inhibiting the calcification process [2]. Increased vascular calcification leads to increased MGP transcription and the production of dephosphorylated-uncarboxylated MGP (dp-ucMGP). Dp-ucMGP is the fully inactive form and is unable to bind calcium or extracellular matrix and therefore sets free into the circulation [3]. Plasma concentration of dp-ucMGP is a better indicator of vascular vitamin K status than other components of MGP [3]. It correlates with an increased risk for cardiovascular mortality and vascular calcification in hemodialysis patients [4,5], as well as in diabetics and heart failure patients [6].

Systemic sclerosis (SSc) is an immune-mediated disease that is characterized by fibrosis of the skin and internal organs and small-vessel vasculopathy. Although rare, SSc has a high mortality and morbidity rate [7]. Atherosclerosis is responsible for 5–8% of deaths in SSc patients [8]. Multiple large cohort studies showed a twofold increased risk for myocardial infarction and stroke and a four- to fivefold increased risk for peripheral vascular disease in SSc patients [9,10,11]. Previously, it was reported that SSc patients have significantly increased carotid intima-media thickness (cIMT) [12,13] and a greater cardiac coronary calcium score (CSS) [14]. Even after adjustment for traditional cardiovascular risk factors, the association between SSc and atherosclerosis persists, suggesting a disease-specific role for inflammation, endothelial and microvascular dysfunction, and subsequent vasculopathy [11]. A better understanding of the underlying mechanism may point to novel targets of therapy. Moreover, there is a clinical need for the identification of inflammatory biomarkers to estimate the risk of cardiovascular disease (CVD) in SSc.

However, classical biomarkers for inflammation, such as C-reactive protein (CRP), are often within the normal range in patients with SSc [15]. Dp-ucMGP appeared to correlate well with CVD in other chronic conditions, such as diabetes mellitus [16]. As dp-ucMGP is predominantly produced within the vascular wall, i.e., the smooth muscle cell, which is also a key cell in the pathogenesis of systemic sclerosis (SSc), we aim to explore the role of dp-ucMGP in patients with SSc.

The aim of this study was to determine dp-ucMGP levels in a well-defined, historical cohort of early-diagnosed SSc patients, to study changes in dp-ucMGP levels during follow-up, and to compare these with an aged-matched Dutch cohort of healthy controls (HCs). For validation, we determine dp-ucMGP in a subset of our prospective cohort. In addition, we studied the incidence of first-ever CVD in the retrospective SSc cohort and investigated if dp-ucMGP as a biomarker predicts CVD and/or all-cause mortality during long-term follow-up and after adjustment for the classical cardiovascular risk factors. Furthermore, we investigated whether dp-ucMGP is associated with the severity of the SSc disease by investigating the predictive value of dp-ucMGP for the presence of interstitial lung disease (ILD) and microvascular injury, i.e., pulmonary arterial hypertension (PAH) and digital ulcers (DU).

## 2. Methods

### 2.1. Study Population and Study Design

The SSc patients evaluated originated from the POEMAS cohort of the Maastricht University Medical Center (MUMC) in the Netherlands, which started in 2005 and aimed to prospectively investigate the prevalence and incidence of pulmonary complications in SSc [17]. Patients were included in the current cohort study if they retrospectively met the 2013 ACR/EULAR criteria for SSc [18], and stored blood samples at −80 °C were still available from baseline inclusion in the POEMAS cohort. Individuals dependent on dialysis three years prior to the sample collection were excluded. In 2020, a follow-up cohort was formed of the SSc patients still being monitored in the MUMC. During this follow-up, a new blood sample was collected, and dp-ucMGP levels were measured in both the baseline and follow-up samples. Additionally, we created another cohort consisting of treatment-naïve SSc patients from our observational prospective cohort study initiated in 2019. In this cohort, dp-ucMGP levels were measured for validation purposes, utilizing the same set of inclusion and exclusion criteria as the original cohort.

Demographic and clinical characteristics were retrieved from medical records. In addition to patient demographics, the collected data contained an auto-antibody profile, SSc subtype, disease manifestations, use of immunosuppressive medication, and date of disease onset, defined as the first non-Raynaud phenomenon. ILD was defined as the presence of ground-glass opacification or fibrosis on high-resolution computed tomography (HRCT). PAH was defined as elevated mean pulmonary arterial pressure (mPAP) ≥ 25 mmHg in rest and supine position measured by right heart catheterization of the pre-capillary type (pulmonary capillary wedge pressure (PCPW) ≤ 15 mmHg) and a pulmonary vascular resistance (PVR) > 3 wood units (WU). The study was approved by the local medical ethics committee and is in accordance with the Declaration of Helsinki.

### 2.2. Measurement of dp-ucMGP

Circulating dp-ucMGP levels were determined in EDTA plasma in a single run by the Laboratory of Coagulation Profile (Maastricht, the Netherlands) using the commercially available IVD CE-marked chemiluminescent InaKtif MGP assay on the IDS-iSYS system (IDS, Boldon, UK) [19,20]. Patient samples were incubated with magnetic particles coated with murine monoclonal antibodies against dpMGP, acridinium-labeled murine monoclonal antibodies against UCMGP, and assay buffer. The magnetic particles were captured using a magnet and washed to remove any unbound analytes. Trigger reagents were added, and the resulting light emitted by the acridinium label was directly proportional to the concentration of dp-ucMGP in the sample. The within-run and total precision of this assay were 0.8–6.2% and 3.0–8.2%, respectively. The assay measuring range was between 300 and 12,000 pmol/L and was found to be linear up to 11,651 pmol/L.

### 2.3. Cardiovascular Events and Risk Factors

The first cardiovascular event occurring during the study period was retrieved. Patients were followed from the date of onset of the disease until they experienced the primary outcome event, death, were lost to follow-up, or the follow-up ended (1 September 2021). Patients with pre-existing CVD were excluded from the analysis of first-ever CVD events.

Traditional cardiovascular risk factors were collected at baseline: cigarette smoking status, diabetes mellitus, antihypertensive treatment, and use of statins. Hypertension was defined as blood pressure over 140/90 mmHg or treatment with anti-hypertensive agents to reach this. Body mass index (BMI) was calculated as the weight in kilograms divided by the height in meters squared. Laboratory test results, including c-reactive protein (CRP), low-density lipoprotein cholesterol (LDLc), and creatinine, were collected at inclusion. CRP was measured using routine turbidimetric analysis on Cobas 8000 (Roche Diagnostics, Almere, The Netherlands); normal values were < 10 mg/L. LDLc was calculated according to the Martin-Hopkins method based on total cholesterol, triglycerides, and HDL cholesterol, all measured by routine turbidimetric analysis on Cobas 8000; normal values are 3.5–4/4 mmol/L. Physician-diagnosed incident cardiovascular events (myocardial infarction, coronary artery disease followed by coronary artery bypass graft (CABG), cerebrovascular accidents including TIA and stroke, and peripheral arterial disease) before and after the baseline date were abstracted. Diagnosis records that specified angina, hemorrhagic strokes, or sudden deaths were not included. Peripheral arterial disease was defined as arterial thromboembolism, occlusion, bypass, or angioplasty of the lower extremities.

### 2.4. Statistical Analysis

Statistical analyses were performed using SPSS Statistics for Windows version 28.0 (IBM, Armonk, NY, USA) and GraphPad version 5.03 (Prism Software, San Diego, CA, USA). Descriptive statistics were calculated for demographic and clinical characteristics. Continuous variables are presented as mean (±standard deviation (SD)) in the case of a normal distribution or median (interquartile range (IQR)) in the case of a non-normal distribution. Categorical variables are presented as numbers (percentages). We summated the total person-years and the number of cardiovascular events in the SSc cohort and then calculated the incidence rate (IR) per 1000 person-years. Groups were compared using the independent *t*-test for normally distributed variables, the Chi-Square test for categorical variables in the case of unpaired comparisons, and the dependent *t*-test or Wilcoxon signed rank test in the case of paired comparisons. Cumulative event-free survival rates at 10 years were computed by the Kaplan–Meier method, and significance was tested with the log-rank test. Event-free survival was defined as death or first-ever cardiovascular morbidity. Multiple group (≥3 groups) comparison was performed using the Kruskal–Wallis test with post hoc analysis and correction for multiple testing. Binary logistic regression models were used to calculate odds ratios (OR) for cardiovascular risk factors and dp-UCMGP for the risk of a cardiovascular event. In addition, ORs were calculated for dp-ucMGP and ILD, or microvascular injury, defined as digital ulcers and/or PAH. Two-sided *p* < 0.05 was considered statistically significant for all analyses.

## 3. Results

### 3.1. Clinical Characteristics of the SSc Cohort and dp-ucMGP Levels Compared with HCs

Eighty-seven SSc patients from the original POEMAS cohort were included for dp-ucMGP analysis in the baseline cohort. This cohort consisted of 64 (73.6%) women, and the mean age at disease onset was 54.5 years (SD ± 19). At inclusion, the median disease duration was 4.0 years (IQR 8.3). The majority of SSc patients had limited skin involvement (88.5%) and anti-centromere antibodies (46.0%). Thirty-one SSc patients (35.6%) were diagnosed with ILD, and six SSc patients had PAH (6.9%). More than half of all SSc patients (59.8%) were treated with immunosuppressive therapy (Table 1).

The median dp-ucMGP level at baseline was 634 pmol/L (IQR 301). This is greatly increased when compared with a mean dp-ucMGP level of <393 pmol/L in 400 age-matched Dutch HCs (*p* < 0.001) [21]. Validation of elevated dp-ucMGP levels in the prospective cohort of SSc patients (*n* = 29) was achieved, with a median level of 589 pmol/L (IQR 370). Clinical characteristics can be found in Appendix A.

### 3.2. dp-ucMGP Levels at Follow-Up in 2020

In 2020, it was found that 33 SSc patients had died during follow-up, and 19 patients were not available for a new blood sample, so a new blood sample was taken from 26 patients. The mean dp-ucMGP level at follow-up has consistently increased by a value of 621 pmol/L (IQR 342) (Figure 1a). No statistical difference was observed between the baseline and follow-up samples (*p* = 0.181; Figure 1b). One notable increase in MGP level was observed in a patient who developed renal ANCA vasculitis with renal function deterioration.

### 3.3. Occurrence and Treatment of Traditional Cardiovascular Risk Factors

Hypertension was detected in 20 (23%) patients, all of whom appeared to be well controlled (Table 2). Cholesterol was analyzed in 68 (78%) patients with an LDL cholesterol level of 3.1 mmol/L (IQR 1.4). Lipid-lowering drugs were used in 22 (25.3%) patients. Only a small number of SSc patients with type 2 diabetes mellitus were present. The median BMI was in the normal range. Data on smoking were available for all patients. There were 18 (20.7%) current smoking patients and 37 (42.5%) patients who smoked in the past.

### 3.4. Cardiovascular Morbidity

Only one patient had evident CVD before the onset of SSc. Nine patients were lost to follow-up; the remaining 78 SSc patients were followed up until the first event of CVD, death, or end of the follow-up period, with a median follow-up time of 11.7 years (IQR 15.5). Clinical characteristics were not significantly different in the follow-up cohort compared with the whole cohort of SSc patients (Table 1).

Twenty-six (33.3%) of seventy-eight SSc patients had a first new cardiovascular event during the study period, of which eighteen (23.1%) were caused by myocardial ischemic disease, four (5.1%) were caused by peripheral vascular disease, and four (5.1%) were strokes. The median time to CVD was 10.5 (IQR 15.2) years from the onset of SSc disease. The incidence rate per 1000 person-years of CVD from onset of SSc during follow-up in this cohort is 29.9.

### 3.5. dp-ucMGP Levels as Predictors for CVD

There were no significant differences between demographic and cardiovascular risk variables at baseline between the 26 SSc patients who developed a first cardiovascular event during follow-up compared with the 52 SSc patients without CVD (Table 2). The dp-ucMGP levels were comparable between patients with and without CVD. Furthermore, disease-specific markers such as type of auto-antibodies, diffuse skin involvement, digital ulcers, CRP, immunosuppressive treatment, ILD, or PAH were not significantly different between patients with and without CVD. Odds ratios for current smoking (OR 3.09; 95%CI 0.86–11.5), sex (OR 2.44; 95%CI 0.89–6.72), and hypertension (OR 2.53; 95%CI 0.86–7.46) tended to predict CVD, but the association did not reach statistical significance (Table 3).

### 3.6. dp-ucMGP and Event-Free Survival in SSc Patients

We next categorized the SSc patients as being above or below the median of dp-ucMGP plasma levels to perform a Kaplan–Meier analysis over the first 10 years after onset of SSc. Kaplan–Meier analysis showed that elevated dp-ucMGP levels (>634 pmol/L) were associated with an increased risk for CVD and/or death during the first 10 years of follow-up (Figure 2; log-rank test: *p* < 0.01).

### 3.7. dp-ucMGP Association with SSc-ILD and Vasculopathy

At baseline, 31 patients (35.6%) were diagnosed with ILD. The univariable regression analysis showed that male sex, diffuse skin involvement, anti-topoisomerase antibodies, and dp-ucMGP levels (as the delta of 300 pmol/L) were significantly predictive for the development of ILD (Table 4). In multivariable regression analysis, anti-topoisomerase antibodies (OR 5.6; 95%CI 1.4–22.0) and dp-ucMGP levels at baseline (OR 1.82; 95%CI 1.01–3.28) were significantly predictive for the development of ILD. No significant association was found for dp-ucMGP and SSc vasculopathy, as determined in patients with digital ulcers (*n* = 40; 46%) and PAH (*n* = 6; 6.9%), nor for calcinosis (*n* = 17; 19.5%).

## 4. Discussion

In this single-center cohort of early SSc patients, plasma dp-ucMGP levels are significantly higher in SSc patients compared with an age-matched control group. These dp-ucMGP levels consistently remain elevated even with longer disease duration. Furthermore, high dp-ucMGP levels early in the disease course already revealed an increased risk for CVD and/or death in SSc, suggesting dp-ucMGP as a novel biomarker to estimate the risk of CVD and/or death in SSc. Moreover, the levels of dp-ucMGP showed to be a significant predictive risk factor for the development of ILD.

Previous studies in patients with diabetic kidney disease reported comparable increased levels of dp-ucMGP [16,22]. It is not clear what causes the increased dp-ucMGP plasma levels in SSc patients. Epidemiological data have repeatedly reported a strong, inverse association between dp-ucMGP and vitamin K status [23,24] in diabetic and hemodialysis patients, suggesting a vascular vitamin K deficiency in SSc patients.

Vitamin K refers to a set of different fat-soluble vitamins occurring as phylloquinone (vitamin K1) or as a series of vitamins termed menaquinones (vitamin K2). The main sources of vitamin K1 are green vegetables and dairy products. Vitamin K2 is mainly considered to be produced by bacterial flora in the intestinal tract [25]. It is reported that gastrointestinal (GI) complications affect up to 90% of SSc patients [26]. The pathogenesis includes fibrosis of enteric connective tissue, vascular damage, smooth muscle cell inflammation and atrophy, and myenteric neural dysfunction due to collagen deposition or autoantibodies. One of the complications of GI involvement in SSc is small intestinal bacterial overgrowth (SIBO) [27]. One may argue that the bacterial overgrowth can cause an increased production of menaquinones [28], but overall SIBO leads to malabsorption of fat-soluble vitamins (A, D, E, and K) in SSc patients [29]. A recent meta-analysis showed a pooled prevalence of SIBO in SSc of 34% (95%CI 27–42%) [30], illustrating that malnutrition is an important phenomenon that can contribute to a vascular vitamin K deficiency leading to high levels of dp-ucMGP. Nonetheless, this study did not examine specific investigations related to gastrointestinal symptoms. Further research is needed to explore the relationship between gastrointestinal symptoms, vitamin K levels, and dp-ucMGP.Vitamin K deficiency is strongly associated with severe arterial calcification [3]. In this regard, it has been proposed that vitamin K therapy has the potential to slow vascular calcification. In a trial of 200 hemodialysis patients, thrice-weekly supplementation for 8 weeks with vitamin K2 resulted in a dose-dependent reduction in plasma dp-ucMGP [24]. Another study in hemodialysis patients showed that vitamin K1 supplementation slowed the average thoracic aortic calcifications as well as the coronary artery calcifications and reduced dp-ucMGP levels [31]. In other populations, like healthy postmenopausal women, vitamin K supplementation decreased dp-ucMGP levels and reduced arterial stiffness [32].

In our SSc cohort, we found a high IR of 29.9 per 1000 person-years for a new cardiovascular event, but traditional cardiovascular risk factors did not predict the development of CVD. An increased risk of CVD, i.e., peripheral vascular disease, myocardial infarction, and stroke, has been reported in SSc patients [10,33,34]. The cause of this elevated risk in SSc patients is still unknown, but it may be related to the increased risk of endothelial dysfunction, microvascular damage, and chronic inflammation, which are all typical features of the disease [35]. Higher levels of interleukin-6 and CRP are positively correlated with atherosclerosis in SSc patients [36,37]. Although CRP is considered a surrogate marker for inflammation, it is often normal in patients with SSc, as our study corroborates. Another possible mechanism is endothelial cell (EC) dysfunction, which is considered the cornerstone of SSc vasculopathy and is induced by innate and adaptive immune responses. Elevated levels of specific biomarkers of endothelial dysfunction related to arginine metabolism, such as asymmetric dimethylarginine (ADMA) or symmetric dimethylarginine (SDMA), were detected in patients with limited cutaneous SSc when compared with control subjects [38]. ADMA metabolism is considered an important mediator of vascular injury and is positively correlated with cIMT in patients with early SSc with a disease duration < 4 years [39]. This implies that ADMA may be associated with accelerated atherosclerosis in the early phases of the disease. EC injury may lead to EC apoptosis. Transdifferentiation of EC via the process of endothelial-mesenchymal transition (endoMT) and of pericytes into profibrotic myofibroblasts may contribute further to vascular wall fibrosis. VSMCs may migrate into the intima, differentiate, and synthesize the matrix of fibrotic vascular lesions. In this manner, endothelial cell dysfunction is further linked to atherosclerosis. In addition, studies have linked higher dp-ucMGP concentrations with other markers of endothelial dysfunction in patients on hemodialysis via brachial artery flow-mediated dilatation [40] and in healthy postmenopausal women via biochemical markers (VCAM, E-selectin) [32]. Finally, it is possible that vitamin K deficiency itself leads to more inflammation and, thus, atherosclerosis, independent of the type of underlying disease. It has been demonstrated that vitamin K can inhibit NLRP3 (NOD-, LRR-, and pyrin domain-containing protein 3) inflammasome activation. NLRP3 is activated by various endogenous danger signals abundantly present in atherosclerotic lesions, such as oxidized low-density lipoproteins and cholesterol crystals [41]. Upon activation, the NLRP3 inflammasome recruits pro-inflammatory cysteinyl aspartate-specific proteinases (caspases) via adaptor-associated speck-like protein (ASC) containing a caspase recruitment domain (CARD), followed by maturation of caspases and secretion of inflammatory cytokines. Vitamin K blocks the interaction needed between ASC and NLRP3 [42]. The accumulation of oxidized low-density lipoproteins is described in SSc due to oxidative stress caused by EC dysfunction. This triggers EC activation, which further amplifies inflammatory processes [33]. Thus, inflammation and EC dysfunction may both contribute to the pathogenesis of atherosclerosis and SSc, where the elevated dp-ucMGP levels may be the link between inflammation, microvascular injury, and atherosclerosis. The question remains how the clinician can reduce the risk of CVD, as this study did not find a reduction in the risk when immunosuppressive therapy was used. 

Different studies suggest a direct association of MGP with inflammation and fibrosis, but precise mechanisms are unknown. One of the proposed mechanisms states that activated MGP creates a complex with bone morphogenetic protein 2 (BMP2), thereby inhibiting BMP2 expression and preventing downstream signals that will lead to endothelial cell (EC) proliferation and VSMC differentiation [43]. In mouse models with experimentally induced colitis, Feng Y et al. showed that mesenchymal stem cell-derived MGP improved the clinical and histopathological severity of colonic inflammation, alleviated T-cell infiltration, and suppressed the production of pro-inflammatory cytokines in colon tissue [44]. Furthermore, MGP is expressed by stimulated monocytes and macrophages at levels similar to IL-1β, showing involvement in the endogenous inflammatory response mechanisms of monocytes and macrophages [45]. In a different mouse model, the targeted removal of the MGP gene in specific cells resulted in the development of severe pulmonary fibrosis. It was discovered that MGP binds to a distinct member of the BMP family, BMP-1, thereby inhibiting its role in promoting the differentiation of endothelial cell-like myofibroblasts through increased transforming growth factor (TGFβ1) signaling. This effectively blocks the impact of EC-like myofibroblast proliferation on pulmonary fibrosis [46].

This study is limited by its retrospective design. However, by taking clear outcome measures with physician-diagnosed incident CVD and a well-defined follow-up till death, first-ever CVD, or date lost to follow-up, this may have been partially overcome. Another limitation is probably the low numbers for the regression analysis. This may explain why no significant differences were shown between the patients with and without CVD. In the multivariable analysis of risk factors for SSc-ILD, skin involvement no longer holds statistical significance. This could be attributed to the inclusion of a substantial population with limited skin involvement in this study (88.5%), which may not be representative of the typical patient population seen in scleroderma clinics. However, it is noteworthy that within the same cohort, dp-ucMGP emerged as a significant predictive risk factor for SSc-ILD. Further research on the relationship between MGP and inflammation, particularly in a larger SSc cohort, is necessary.

Despite these limitations, the strength of our study is that it is the first, to our knowledge, that determines dp-ucMGP levels in a group of SSc patients and links these to cardiovascular events and severity of disease. Second, the cohort consists of patients with early-diagnosed SSc, including only one patient with pre-existing CVD. Furthermore, elevated dp-ucMGP levels were confirmed in a second prospective cohort of SSc patients. The retrosepective study had a long-term follow-up, allowing a reliable representation of first-ever CVD and/or death. The significantly negative effect of higher levels of dp-ucMGP on event-free survival in SSc patients indicates that dp-ucMGP is a new biomarker for disease.

In conclusion, this study showed increased dp-ucMGP levels in SSc patients compared with age-matched controls. We confirm the high risk of CVD in SSc patients; however, traditional cardiovascular risk factors did not predict the development of CVD. In contrast, high dp-ucMGP levels revealed an increased risk for CVD and/or death in SSc and additionally showed to be a significant predictive risk factor for the development of ILD. It is still unclear what caused the increased dp-ucMGP levels in SSc patients, but given the strong, inverse association between dp-ucMGP and vitamin K status, a vitamin K deficiency is proposed. Whether this is caused by malabsorption, microvascular injury, or inflammation requires further research.

## Figures and Tables

**Figure 1 diagnostics-13-03526-f001:**
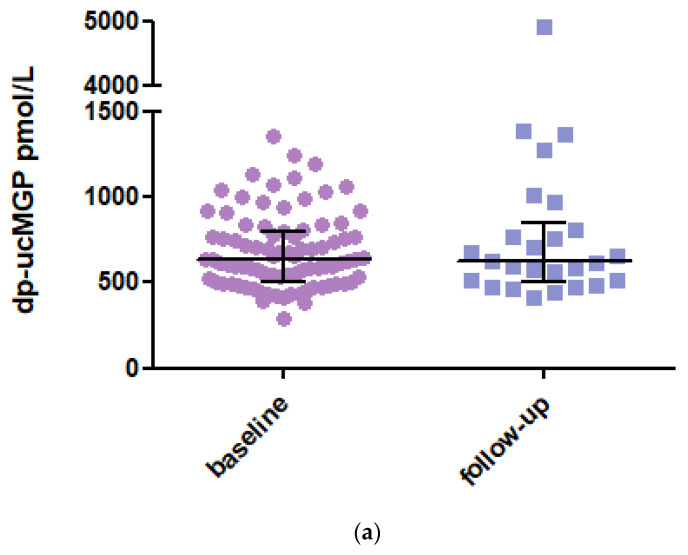
(**a**) dp-ucMGP levels at baseline and in 2020 at follow-up. Data are presented as median with IQR. Dp-ucMGP levels (pmol/L) at baseline (*n* = 87) and follow-up (*n* = 26). (**b**) Paired analysis of the dp-ucMGP levels of the 26 SSc patients between baseline and follow-up. Data are presented as paired values of dp-ucMGP levels at baseline and follow-up. Dp-ucMGP: Dephosphorylated and uncarboxylated Matrix Gla Protein.

**Figure 2 diagnostics-13-03526-f002:**
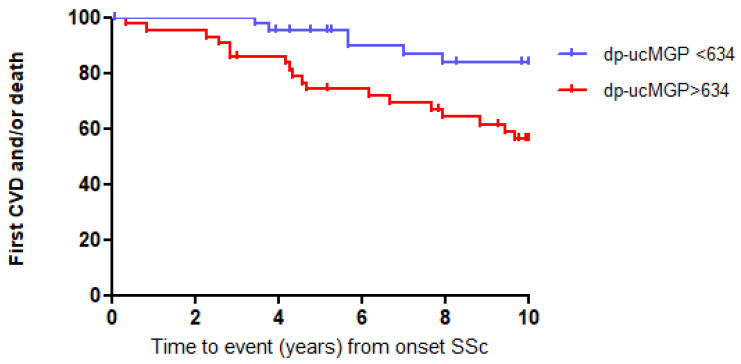
High dp-ucMGP levels predict event-free survival in SSc patients, log-rank test: *p* < 0.01.

**Table 1 diagnostics-13-03526-t001:** Clinical characteristics of the whole SSc cohort (*n* = 87) and of SSc patients who were followed up until the first event of CVD, death, or end of follow-up period (*n* = 78).

Clinical Features	SSc Cohort *n* = 87	SSc Follow-Up *n* = 78	*p*
Age at inclusion, y	57.9 (11.5)	58.2 (11.7)	0.969
Age at disease onset ^¥^, y	54.5 (19.0)	57.1 (16.8)	0.631
Female sex	64 (73.6)	55 (70.5)	0.760
Disease duration, y	4.0 (8.3)	3.2 (7.4)	0.601
Duration follow-up ^$^ y	10.7 (8.8)	11.8 (10.3)	0.859
Skin involvement			0.988
Limited	77 (88.5)	69 (88.5)	
Diffuse	7 (8.0)	6 (7.7)	
Sine sclerosis	3 (3.4)	3 (3.8)	
Antibody			1.000
Centromere	40 (46.0)	37 (47.4)	
Topoisomerase I	18 (20.7)	18 (23.1)	
RNA polymerase III	2 (2.3)	2 (2.6)	
PM-Scl	1 (1.1)	1 (1.3)	
ANA only	23 (26.4)	16 (20.5)	
None	3 (3.4)	4 (5.1)	
Calcinosis	17 (19.5)	15 (19.2)	0.906
Digital ulcers	40 (46.0)	34 (43.6)	0.758
ILD	31 (35.6)	27 (34.6)	0.896
Pulmonary hypertension	6 (6.9)	4 (5.1)	0.906
Immunosuppressive therapy	52 (59.8)	49 (62.8)	0.806
Laboratory parameters			
Creatinine, umol/L	74 (23)	73 (21)	0.943
CRP mg/L	4.1 (5.1)	5.2 (7.1)	0.947
sIL2r, U/mL	338 (321) *n* = 43	338 (321) *n* = 39	0.710
dp-ucMGP, pmol/L	634 (301)	677 (341)	0.964

Data are presented as a number (%), mean ± SD, or median (IQR). ANA: anti-nuclear antibodies; CRP: c-reactive protein. Dp-ucMGP: Dephosphorylated and uncarboxylated Matrix Gla Protein; ILD: interstitial lung disease; IQR: interquartile range. sIL2r: soluble interleukin 2 receptor. ¥ Defined as the date of the first non-Raynaud’s phenomenon symptom. $ Defined as duration from date of disease onset until first CVD, death, loss of follow-up, or end of study period (1 September 2021). Reference ranges for creatinine are 60–115 umol/L, CRP < 10 mg/L, and sIL2r < 600 U/mL.

**Table 2 diagnostics-13-03526-t002:** Cardiovascular risk factors and dp-ucMGP at inclusion in the whole SSc cohort and in SSc patients with and without CVD.

	SSc Cohort *n* = 87	Cohort CVD *n* = 26	No CVD *n* = 52	*p*
Female sex	64 (73.6)	15 (57.7)	40 (76.9)	0.079
Age at inclusion, y	57.9 (11.5)	60.0 (12.5)	56.8 (11.2)	0.683
Hypertension	20 (23.0)	9 (34.6)	9 (17.3)	0.087
Good control	20 (23.0)	9 (34.6)	9 (17.3)	
Diabetes mellitus	5 (5.7)	2 (7.7)	3 (5.8)	0.744
Smoking				0.212
Current	18 (20.7)	8 (30.8)	8 (15.4)	
Former	37 (42.5)	11 (42.3)	22 (42.3)	
No	32 (36.8)	7 (26.9)	22 (42.3)	
LDL cholesterol, mmol/L	3.1 (1.4)	3.1 (1.7)	3.1 (1.3)	0.919
Lipid-lowering drug at baseline	22 (25.3)	10 (38.5)	10 (19.2)	0.067
BMI, kg/m^2^	23.9 (5.5)	24.5 (5.6)	25.6 (7.4)	0.849
VKA use	4 (4.6)	1 (3.8)	3 (5.8)	0.717
dp-ucMGP, pmol/L	634 (301)	655 (401)	634 (343)	0.483

Data are presented as number (%), mean ± SD or median (IQR). BMI: body mass index; Dp-ucMGP: Dephosphorylated and uncarboxylated Matrix Gla Protein; IQR: interquartile range. LDL: low-density lipoprotein; VKA: vitamin K antagonist. LDL cholesterol reference range: 3.5–4.4 mmol/L. Hypertension was defined as blood pressure over 140/90 mmHg at baseline or treatment with anti-hypertensive agents.

**Table 3 diagnostics-13-03526-t003:** Impact of baseline factors on the risk of cardiovascular events during follow-up.

	POEMAS Cohort *n* = 78 OR (95% CI)	*p*
Age at inclusion	1.03 (0.98–1.07)	0.253
Sex	2.44 (0.89–6.72)	0.083
Hypertension	2.53 (0.86–7.46)	0.093
Good control		
Diabetes mellitus	1.36 (0.21–8.70)	0.745
Smoking		
Current	3.09 (0.86–11.5)	0.084
Former	1.59 (0.51–4.80)	0.428
LDL cholesterol, mmol/L	1.10 (0.65–1.86)	0.727
Lipid-lowering drug at baseline	2.63 (0.92–7.50)	0.071
BMI, kg/m^2^	0.995 (0.90–1.1)	0.927
dp-ucMGP, pmol/L	1.001 (0.999–1.003)	0.219

BMI: body mass index; Dp-ucMGP: Dephosphorylated and uncarboxylated Matrix Gla Protein; LDL: low-density lipoprotein; OR: odds ratio. Hypertension was defined as blood pressure over 140/90 mmHg at baseline or treatment with anti-hypertensive agents. LDL cholesterol reference range: 3.5–4.4 mmol/L.

**Table 4 diagnostics-13-03526-t004:** Impact of baseline factors on the risk for ILD at baseline: univariable and multivariable analysis in POEMAS cohort *n* = 87. The effect of dp-ucMGP is shown as the delta of 300 pmol/L.

	Univariable Analysis OR (95% CI)	Multivariable Analysis OR (95% CI)
Age at inclusion	1.02 (0.98–1.06)	
Sex	3.25 (1.21–8.74) *	1.98 (0.59–6.61)
Smoking		
Current	1.20 (0.35–4.17)	
Former	1.5 (0.56–4.06)	
Skin involvement		
Diffuse	13 (1.48–114.0) *	6.74 (0.56–81.42)
dp-ucMGP, pmol/L	2.46 (1.82–4.43) **	1.82 (1.01–3.28) *
CRP	1.01 (0.99–1.04)	
sIL2r	1.00 (0.998–1.002)	
Anti-topoisomerase	10.5 (3.0–36.2) ***	5.6 (1.4–22.0) *

CRP: c reactive protein; Dp-ucMGP: Dephosphorylated and uncarboxylated Matrix Gla Protein; sIL2r: soluble interleukin 2 receptor; OR: odss ratio. Reference values: CRP < 10 mg/L, sIL2r < 600 U/mL. * *p* ≤ 0.05; ** *p* ≤ 0.01; *** *p* ≤ 0.001.

## Data Availability

The data presented in this study are available on request from the corresponding author. The data are not publicly available due to ethical restrictions.

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
