# Peer review of "Plasma Dephosphorylated-Uncarboxylated Matrix Gla-Protein in Systemic Sclerosis Patients: Biomarker Potential for Vascular Calcification and Inflammation"

_diagnostics, 2023, doi:10.3390/diagnostics13233526_

Round 1

Reviewer 1 Report

Comments and Suggestions for Authors

The authors provided an interesting article entitled „Plasma dephosphorylated-uncarboxylated Matrix Gla-Protein  in systemic sclerosis patients: biomarker potential for vascular 3 calcification and inflammation“ regarding a dephosphorylated-uncarboxylated Matrix Gla-Protein as a novel biomarker in systemic sclerosis patients. However, here are some modifications and recommendations.

Please, explain how CRP and LDLc were determined.

Table 1. Please arrange abbreviations alphabetically (same for all Tables). Further, it could be split into 2 tables -  the second one could be with creatinine, CRP, sIL2R, and dp-ucMGP values.

Figure 1b could be presented with a different graph, perhaps more simple.

Table 2. It is not necessary for the mean and median to be in the table (actual words), it is enough to write it down in the legend (data are present as...). Same for Tables 1 and 3.

Diabetes mellitus detected among patients, which one type?

Why other lipid parameters were not determined?

Author Response

We would like to thank the reviewers for the positive feedback and helpful comments for correction and modification.

Reviewer comments:

Please, explain how CRP and LDLc were determined.

  • This is added in the Methods section.

Table 1. Please arrange abbreviations alphabetically (same for all Tables). Further, it could be split into 2 tables -  the second one could be with creatinine, CRP, sIL2R, and dp-ucMGP values.

  • Abbreviations have now been put in alphabetical order in all tables. We opted not to include a second table. Instead, we arranged the laboratory diagnostics under a distinct heading, with the outcome parameter, dp-ucMGP, positioned at the end. This adjustment enhances clarity in our manuscript.

Figure 1b could be presented with a different graph, perhaps more simple.

  • This has been adjusted.

Table 2. It is not necessary for the mean and median to be in the table (actual words), it is enough to write it down in the legend (data are present as...). Same for Tables 1 and 3.

  • This is corrected in all tables.

Diabetes mellitus detected among patients, which one type?

  • There were only patients with type 2 diabetes mellitus present in the cohort. The type is added in the text.

Why other lipid parameters were not determined?

  • In addition to LDL cholesterol, the total cholesterol and HDL cholesterol have also been determined. However, the primary cholesterol parameter considered a risk factor for cardiovascular diseases is the LDL cholesterol level in the blood. Therefore, it is presented alongside the other risk factors instead of total cholesterol and HDL cholesterol.

Reviewer 2 Report

Comments and Suggestions for Authors

This interesting study investigates the association between another one vascular marker (Matrix Gla-protein (MGP) and cardiovascular disease in SSc patients. This relatively new and unknown marker – at least at the rheumatology field - was elevated in patients with SSc and was found to predict CVD events in this population.

I have some comments.

A better rationale for the study is required as the data for this marker especially across the spectrum of systemic diseases is limited.

The majority of the study population (85%) consisted of patients with limited disease, which apparently is not representative of the average population attending scleroderma clinics. This should be commented and considered on the multivariate analysis.

Whistle one of the possible explanations provided is associated with malnutrition and malabsorption observed in SSc patients, the percentage of patients presenting with GI involvement in the current study is not mentioned and not included as a confounder in the analysis.

The discussion needs to be more focused on the pathophysiology of atherosclerosis in SSc. The authors discuss very extensively potential mechanisms including the association between Vitamin K and NLRP3 activation highlighting the autoinflammatory component in the development of vascular disease not only in SSc (Rheumatol Int. 2018 Aug;38(8):1345-1354) but also in general. On the other hand the role of inflammation in the development of atherosclerosis in SSc is less pronounced compared with other systemic disease such as Rheumatoid Arthritis and other SSc-related mechanisms such as microvasculopathy have an important effect on CVD disease. To lend more support the study did not demonstrate any association between dp-ucMGP levels and inflammatory markers (CRP is normal in the study population) or reduction in CVD risk with the administration of immunosuppressive therapy. In this regard a more critical approach in endothelial dysfunction and CVD risk focused on SSc should be presented in the discussion including and other potential markers (Clin Exp Rheumatol 2021:39(131):57–65. Mayo Clin Proc. 2020 Jul;95(7):1369-1378. Clin Rheumatol. 2023 Apr;42(4):1077-1085).

The data regarding immunosuppressive treatment and the effect on the study outcomes are lacking. Is there any link between the type of treatment (vasodilators, steroids, conventional DMARDs, biologics, antifibrotics) with the levels of dp-ucMGP?

One of the most important results of the analysis namely the association between dp-ucMGP and pulmonary fibrosis is not discussed by the authors. Pulmonary fibrosis is considered one of the leading causes of death in this population and the higher mortality demonstrated in SSc individuals with dp-ucMGP levels (>634 pmol/L) may also be attributed to the presence of ILD in parallel with CVD events. More importantly dp-ucMGP may have a role in Transforming growth factor signaling and subsequently to the development of fibrosis. This point with relevant clinical implications should be discussed (ERJ Open Res 2023; 9: 00487-2022. Eur Respir J 2023 Sep 28;62(3):2301329, Cell Mol Gastroenterol Hepatol. 2023 Oct 12:S2352-345X(23)00172-8)

Author Response

We would like to thank the reviewer for the positive feedback and helpful comments for correction and modification.

  • Reviewer 2 comments:

    A better rationale for the study is required as the data for this marker especially across the spectrum of systemic diseases is limited.

    • We appreciate the reviewer for providing this critical comment. We have included an additional paragraph in the introduction to better clarify the study's rationale.

    The majority of the study population (85%) consisted of patients with limited disease, which apparently is not representative of the average population attending scleroderma clinics. This should be commented and considered on the multivariate analysis.

    • We have included this limitation to the discussion. The limited skin involvement did not have predictive value for the occurrence of ILD. In the multivariate analysis, skin involvement (diffuse or limited) was not statistically significant (shown in Table 4).

    Whilst one of the possible explanations provided is associated with malnutrition and malabsorption observed in SSc patients, the percentage of patients presenting with GI involvement in the current study is not mentioned and not included as a confounder in the analysis.

    • The demographic and clinical characteristics of the SSc patients were collected retrospectively. Because gastrointestinal (GI) involvement is often a subjective matter, and standardized questionnaires like the UCLA GIT 2.0 score were not used in the clinic, it was decided not to mention the GI manifestations in the manuscript. Therefore, the choice was made to describe the percentage of gastrointestinal complaints in general. Since these are common and have even been reported in populations up to 90%, they are also less relevant as potential confounders. We added a small comment in the discussion.

    The discussion needs to be more focused on the pathophysiology of atherosclerosis in SSc

    • We would like to thank the reviewer for pointing out valuable papers on the role of endothelial dysfunction in the development of atherosclerosis. We have incorporated this into the discussion.

    The data regarding immunosuppressive treatment and the effect on the study outcomes are lacking. Is there any link between the type of treatment (vasodilators, steroids, conventional DMARDs, biologics, antifibrotics) with the levels of dp-ucMGP?

    • At the time of baseline sampling (2005-2009), antifibrotics and biologics were not yet available for the treatment of SSc. We identified from the medical records the use of corticosteroids alone, or DMARDs either with or without corticosteroids. Data on vasodilators were incompletely available to draw reliable conclusions. We did not observe any impact of medication on the baseline dp-ucMGP levels. The indication for and the duration and type of immunosuppression was variable and not standardized and therefore reliable conclusions cannot be drawn from our data. Patients were always treated using what was considered optimal clinical care by their treating physician. However, in future prospective studies the use of medication has indeed to be taken into account.

    One of the most important results of the analysis namely the association between dp-ucMGP and pulmonary fibrosis is not discussed by the authors.

    • We would like to thank the reviewer for the informative paper on the associations between MGP deficiency and pulmonary fibrosis. We have incorporated the explanation of the findings from the mouse model into the discussion section.

Round 2

Reviewer 2 Report

Comments and Suggestions for Authors

NO COMMENTS - AUTHORS ADRESSED THE COMMENTS SATISFACTORILY